**Current potential of CH4 emission estimates using TROPOMI in the Middle East**
Mengyao Liu[1]*, Ronald van der A[1], Michiel van Weele[1], Lotte Bryan[1,2], Henk Eskes[1],
Pepijn Veefkind[1, 2], Yongxue Liu[3], Xiaojuan Lin[1,4], Vincent Huijnen[1], Jos de Laat[1],
Jieying Ding[1]
[1] KNMI, Royal Netherlands Meteorological Institute, De Bilt, The Netherlands
[2] Delft University of Technology, Delft, The Netherlands
[3] School of Geographic and Oceanographic Sciences, Nanjing University, Nanjing,
China
[4] Department of Earth System Science, Ministry of Education Key Laboratory for Earth
System Modeling, Tsinghua University, Beijing, China
* Correspondence to: Mengyao Liu (mengyao.liu@knmi.nl)
**Abstract**
An improved divergence method has been developed to estimate annual methane ($CH_4$)
emissions from TROPOspheric Monitoring Instrument (TROPOMI) observations. It
has been applied to the period of 2018 to 2021 over the Middle East, where the
orography is complicated, and the mean mixing ratio of methane ($XCH_4$) might be
affected by albedos or aerosols over some locations. To adapt to extreme changes of
terrain over mountains or coasts, winds are used with their divergent part removed. A
temporal filter is introduced to identify highly variable emissions and further exclude
fake sources caused by retrieval artifacts. We compare our results to widely used
bottom-up anthropogenic emission inventories: Emissions Database for Global
Atmospheric Research (EDGAR), Community Emissions Data System (CEDS) and
Global Fuel Exploitation Inventory (GFEI) over several regions representing various
types of sources. The $NO_X$ emissions from EDGAR and Daily Emissions Constrained
by Satellite Observations (DECSO), and the industrial heat sources identified by Visible
Infrared Imaging Radiometer Suite (VIIRS) are further used to better understand our
resulting methane emissions. Our results indicate possibly large underestimations of
methane emissions in metropolises like Tehran (up to 50%) and Isfahan (up to 70%) in
Iran. The derived annual methane emissions from oil/gas production near the Caspian
Sea in Turkmenistan are comparable to GEFI but more than two times higher than
EDGAR and CEDS in 2019. Large discrepancies of distribution of methane sources in

Riyadh and its surrounding areas are found between EDGAR, CEDS, GFEI and our emissions. The methane emission from oil/gas production in the east to Riyadh seems to be largely overestimated by EDGAR and CEDS, while our estimates, and also GFEI and DECSO $NO_X$ indicate much lower emissions from industry activities. On the other hand, regions like Iran, Iraq, and Oman are dominated by sources from oil and gas exploitation that probably includes more irregular releases of methane, with the result that our estimates, that include only invariable sources, are lower than the bottom-up emission inventories.

## 1 Introduction

Methane ($CH_4$) is the second most important greenhouse gas of which the abundance kept increasing in the last decades (Turner et al., 2019; Saunois et al., 2020; Eyring et al., 2021), with a short-term stable concentration level between the years 2000 and 2006 (Dlugokencky et al., 2009; Rigby et al., 2008). The relatively short lifetime of about a decade makes $CH_4$ emissions a short-term target for mitigating climate change. The TROPOspheric Monitoring Instrument (TROPOMI) on board the Sentinel 5 Precursor (S5-P) satellite provides an opportunity to measure $CH_4$ globally at a high resolution of $7 \times 7$ km² since its launch in October 2017 (upgraded to $5.5 \times 7$ km² in August 2019) (Veefkind et al., 2012; Lorente et al., 2021). Previous studies have demonstrated the capability of TROPOMI to identify big $CH_4$ emitters (e.g., leakages from pipelines) through detecting large anomalies or to derive regional emission fields (de Gouw et al., 2020; Pandey et al., 2019; Zhang et al., 2020; Chen et al., 2023).

However, using observations from TROPOMI to quantify emissions are also facing challenges. On the one hand, some sources are located near the coast or in places with complex topography, where satellite observations are often of reduced quality. The observations of TROPOMI $CH_4$ contain uncertainties from retrieval assumptions for surface albedo, aerosols, and the sun-glint model over the ocean. On the other hand, the characteristics of the various sources are poorly understood. For instance, constant emitting sources from landfills *versus* intermittent leakage of oil/gas, makes it difficult to quantify their emissions (Varon, 2021).

The Middle East is one of the strong $CH_4$-emitting regions in the world (Chen et al., 2023). Nevertheless, these emissions are particularly challenging to be quantified because of the aspects aforementioned. Lauvaux et al. (2022) found fewer detections of ultra-emitters (>25 kg/hour) in Middle Eastern countries like Iraq, Saudi Arabia than other hot-spot regions like the U.S. from TROPOMI observations. Chen et al., (2023) also revealed large discrepancies between a priori and posterior emission inventory derived from satellites over the Middle East.

In this study, we present an improved divergence method (Beirle et al., 2019, 2023; Liu et al., 2021; Sun et al., 2022; Veefkind., 2023) to quantify the emissions of $CH_4$ over the Middle East from 2018 to 2021 on a grid of 0.2° from TROPOMI retrieved $XCH_4$ by using the latest version of the scientific retrieval product (TROPOMI/WFMD v1.8) from the University of Bremen (Schneising et al., 2023). This inversion algorithm is based on the mass balance theory and is unique because of its speed and no need for a priori knowledge of the sources. The wind divergence was first removed from the daily wind fields to better adapt to the complicated orography in the Middle East, and a temporal filter was developed in this study to exclude incorrect sources caused by retrieval issues, respectively. For an area without influence from retrieval issues (e.g., albedo), the persistence of sources can be further tested by the temporal filter.

Before calculating the divergence, we exclude contaminated pixels with a high aerosol optical depth (AOD) using daily MODIS AOD observations and the global hourly Atmospheric Composition Reanalysis 4 (EAC4) dataset. To a grid cell that shows a strong spatial correlation between the divergence and its corresponding background divergence, a posterior correction is applied to remove the contribution from the inhomogeneous background. The final results are further compared to the total anthropogenic $CH_4$ emissions from Emissions Database for Global Atmospheric Research (EDGAR) v7.0 (Crippa et al., 2022) and CEDS v_2021_04_21 (O'Rourke et al., 2021). Other auxiliary datasets, such as the methane emissions from the fuel exploitation predicted by GEFI v2 (Scarpelli et al., 2019) and total anthropogenic $NO_X$ emissions from EDGAR v6.1 and DECSO v6.2 (van der A et al., 2024; Ding et al., 2020; Mijling and van der A, 2012) are used for a better interpretation of our results.

**2 Data and Methodology**

*2.1 Selection of reliable TROPOMI $XCH_4$ data*

This study used the latest TROPOMI WFM-DOAS (TROPOMI/WFMD v1.8) XCH4 product (Schneising et al., 2023). Quality filters were applied to reduce the size of a daily $XCH_4$ file before making it available to the public. Thus, the daily files contain only the pixels that had passed the quality check. In version 1.8, a de-striping filter has been applied to each orbit.

The TROPOMI/WFMD algorithm has been designed for clear-sky scenes with minor scattering by aerosols and optically thin clouds (i.e., cirrus). Still, a few pixels could contain high aerosol loadings (MODIS AOD at 550 *nm* $\geq$ 0.75, Fig. 1. d–f v.s. a-c), leading to biased high $XCH_4$. We here use the daily observation of 10 km MODIS/Aqua AOD data at 550 *nm*, which has a similar overpass time as TROPOMI, to estimate the AOD values for pixels of TROPOMI. The pixels with AOD $\geq$ 0.75 are filtered, and 1.7% of pixels in 2019 are excluded with this criterion in the domain of 10–40N°, 20–50E°. Admittedly, not every TROPOMI pixel has a collocated MODIS AOD observation. Thus, we used the global hourly EAC4 dataset combined with MODIS daily observations to ensure every pixel of TROPOMI has an AOD estimate to reduce the systematic biases caused by high aerosol loadings while maintaining as many pixels as possible. The details about obtaining an AOD value for each pixel can be found in Part A of the Supplementary Information (SI).

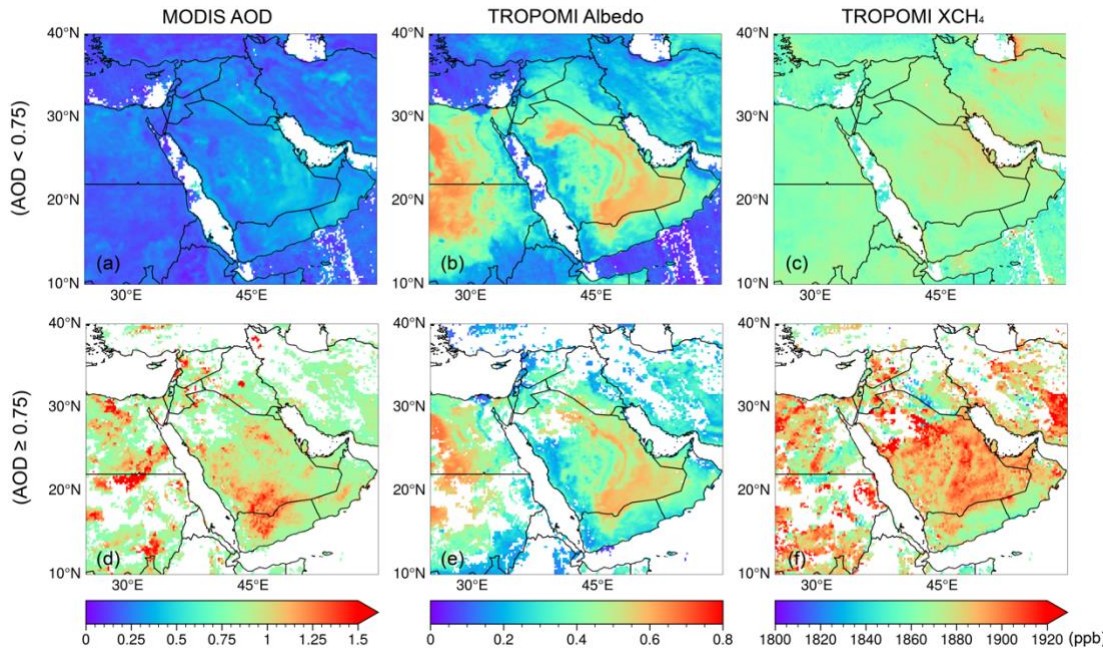

**Figure 1.** Annual mean of (a) MODIS AOD, (b) albedo in TROPOMI XCH4 retrieval and (c) TROPOMI XCH4 on a grid of 0.2° in 2019, which are the average of pixels with AOD < 0.75. (d)-(f) are similar to (a)-(c) but based on the pixels with AOD ≥ 0.75. Only pixels with available MODIS AOD are used to generate the maps shown here.

Another aspect that is addressed is the distinction between land and water bodies, especially over the coastlines. TROPOMI use different retrieval strategies for data over land and ocean. The retrievals over ocean are only available in sun glint mode. We find the data over ocean can be quite noisy. Furthermore, the data continuous from land to ocean are checked. We selected pixels locating at several 1° ×1° areas covering half land and half ocean at the coastlines of Oman, Yemen and along the Red Sea. We found there are not many differences between pixels over land and ocean (see Figure S1 in SI). Therefore, we built a water-land mask at the same spatial resolution as our emission data (0.2° ×0.2°) based on Global Land Cover Characterization (GLCC) of the United States Geological Survey (USGS) (United States Geological Survey, 2018a, b) to distinguish water, land and the coast (transition grids from land to water). Only grid cells that are marked as land and coast are used to build the regional background and are used to calculate the daily divergence.

*2.2 Methane bottom-up emission inventories and auxiliary emission datasets*

In this study, EDGAR v7.0 is mainly used to evaluate the result of the derived methane emissions because it covers the whole period of our study. EDGARv7.0 provides estimates for emissions of the three main greenhouse gases ($CO_2$, $CH_4$, $N_2O$) per sector and country from 1970 to 2021 on a grid of 0.1°. The activity data for non-$CO_2$ emissions are primarily based on the World Energy Balances data (2021) of the IEA.

The activity data for certain sectors are further modified by other updated datasets. For
example, International Fertiliser Association (IFA) and Gas Flaring Reduction
Partnership (GGFR)/U.S. National Oceanic and Atmospheric Administration (NOAA),
United Nations Framework Convention on Climate Change (UNFCCC) and World
Steel Association (worldsteel) recent statistics are used for activity data of energy-
related sectors, and agricultural sectors are further modified by FAO (2021). In addition,
the latest version (v_2021_04_21) of CEDS and the Global Fuel Exploitation Inventory
(GFEI v2) are also used for comparisons in specific years. CEDS v_2021_04_21
consists of CMIP6 historical anthropogenic emissions data from 1980 - 2019 on a grid
of 0.5°. The 0.5° data was further downscaled to 0.1° using 0.1° proxy data from
EDGAR v5.0 emission grids (O'Rourke et al., 2021). GFEI v2 allocates methane
emissions from oil, gas, and coal to a grid of 0.1° by using the national emissions
reported by individual countries to UNFCCC and assign them to infrastructure
locations. GFEI v2 inventory is available for 2019 and presents an update of GFEI v1
which was made for 2016 (Scarpelli, et al., 2021).
Despite the fact that the three above-mentioned inventories have assembled various
information from recent statistics, emissions in the Middle East are still uncertain and
show large discrepancies because of the lack of reports from the industrial facilities. To
validate the sources not reported in bottom-up inventories, target-mode instruments
with very high spatial resolution (pixels < 60m) (e.g., GHGSat, PRISMA, EMIT) are
widely used to pinpoint individual sources and reveal their characteristics. NASA's
Earth Surface Mineral Dust Source Investigation (EMIT) mission was launched in 2020
and methane plumes are recorded since $10^{th}$ August 2022 (Source:
https://earth.jpl.nasa.gov/emit/data/data-portal/Greenhouse-Gases/). It uses an
advanced imaging spectrometer instrument that measures a spectrum for every point in
the image. The high-confidence research grade methane plume complexes from point
source emitters are released as they are identified (Brodrick et al., 2023). In addition,
$NO_X$ emissions and gas flaring data are often used to analyze the emission of methane,
especially for the energy-related sources. Thus, we further used $NO_X$ emissions and
industrial heat sources identified by VIIRS (Liu et al., 2018) to better understand the
derived methane emissions. The latest $NO_X$ emissions from EDGAR (v6.1, the most
recent year is 2018) and the top-down $NO_X$ emission inventory from TROPOMI,
DECSO (van der A et al., 2023; Ding et al., 2020), are used to assess uncertainties of
various emission inventories. For clarity, we combined the source sectors of methane
in EDGAR and CEDS, and the sectors of $NO_X$ in EDGAR into two categories: energy
and others. The sectors for each category are listed in Table-1.

**Table 1. Sectors of CH$_4$ and NO$_X$ used in this study based on EDGAR**

| Sector / Species | Energy | Others |
|---|---|---|
| [1]EDGAR v7.0 CH$_4$ | 1, Power industry (1A1a)<br>2, Refineries and transformation industry (1A1b+1A1ci+1A1cii+1A5biii+1B1b+1B2aiii6+1B2biii3+1B1c)<br>3, Combustion for manufacturing (1A2)<br>4, Fuel exploitation (1B1a+1B2aiii2+1B2aiii3+1B2bi+1B2bii)<br>5, Chemistry process (2B)<br>6, Energy for building (1A4 +1A5)<br>7, Iron and steel production (2C2)<br>8, Fossil fuel fires (5B) | **Transportation**<br>1, Aviation (1A3a)<br>2, Railways, pipelines, off-road transport (1A3c+1A3e)<br>3, Shipping (1A3d)<br>**Agricultural**<br>1, Manure management (3A2)<br>2, Agricultural soils (3C2+3C3+3C4+3C7)<br>3, Enteric fermentation (3A1)<br>**Waste**<br>1, Agricultural waste burning (3C1b)<br>2, Solid waste incineration (4C)<br>3, Solid waste landfills (4A+4B) |
| [2]CEDS v_2021_04_21 CH$_4$ | 1, Energy<br>2, Industrial<br>3, Solvents production and application | 0, Agriculture<br>1, Transportation<br>2, Residential, commercial, other<br>6, Waste<br>7, International shipping |
| EDGAR v6.1 NO$_X$ | 1, Power industry (1A1a)<br>2, Refineries and transformation industry (1A1b+1A1ci+1A1cii+1A5biii+1B1b+1B2aiii6+1B2biii3+1B1c)<br>3, Combustion for manufacturing (1A2)<br>4, Fuel exploitation (1B1a+1B2aiii2+1B2aiii3+1B2bi+1B2bii)<br>5, Chemistry process (2B)<br>6, Energy for building (1A4 +1A5)<br>7, Iron and steel production (2C2)<br>8, Fossil fuel fires (5B)<br>9, Non-ferrous metals production (2C3-C5)<br>10, Food and paper (2H) | **Transportation**<br>1, Aviation (1A3a)<br>2, Railways, pipelines, off-road transport (1A3c+1A3e)<br>3, Shipping (1A3d)<br>**Agricultural**<br>1, Manure management (3A2)<br>2, Agricultural soils (3C2+3C3+3C4+3C7)<br>**Waste**<br>1, Agricultural waste burning (3C1b)<br>2, Solid waste incineration (4C) |

[1]The codes in parentheses are based on IPCC 2006 used by EDGAR v7.0 to generate each sector.
[2]CEDS provides monthly sectoral methane emissions, in which the category is illustrated by the number.

*2.3 Divergence calculation*

The basic methodology has been described in Liu et al. (2021). Here, we have improved the procedure to estimate CH4 emissions from TROPOMI retrieved XCH4 consisting of three steps: (1) The use of daily MODIS/Aqua AOD 10 km L2 dataset (v6.1) and daily CAMS gridded AOD re-analysis data to filter unreliable retrievals of TROPOMI XCH4. (2) Derive the enhancements of XCH4 in the PBL ($XCH_4^{PBL}$) and non-divergent winds from ERA5 wind dataset, which are then used to calculate the spatial divergence and the preliminary methane emission. (3) Apply a posterior spatial correction to subtract the contribution of the residue of the regional background, and identify possible false sources by using a temporal filter.

Our method to estimate the preliminary methane emission $E'$ over a certain period is based on the divergence method described by Beirle et al. (2019) for NOx emissions and specifically for methane by Liu et al. (2021):

$$E' = \overline{D_d^S} = \overline{\nabla(\ (X_d^{PBL} - X_d^B) \ \times \ A_d^{PBL}\ \vec{w})} \quad (1)$$

where $D_d^S$ is the daily divergence of a source. $X_d^{PBL}$ is the daily XCH4 in the Planetary Boundary Layer (PBL) that is calculated by subtracting the vertical column of methane above the PBL from the TROPOMI observations. Estimating the XCH4 in lower atmosphere is quite important since the enhancement due to the transport in the upper atmosphere is irrelevant to the ground emissions. This vertical column above the PBL, is based on the  model results of EAC4 of CAMS at a relative high spatial resolution, 0.75° horizontally and 60 layers vertically (Inness et al., 2019), with methane serving as a background species for chemical reactions. This EAC4 model run contains no *a priori* CH4 emissions. Thus, the spatial distribution of CH4 is mainly driven by transport and orography, which will be subtracted from TROPOMI observations to estimate the PBL concentration of CH4. It is important to note that the total dry air column from the EAC4 dataset is constrained by the TROPOMI retrieval for each pixel, which guarantees the mass conservation. We fixed the PBLH at 500 meters above the ground considering the PBLH from the reanalysis dataset has large uncertainties and is occasionally too shallow (Guo et al., 2021). The favorable height is suggested to be 500-700 meters above the ground considering the systematic difference between EAC4 dataset and TROPOMI observations (Liu et al., 2021). $X_d^B$ is the regional background of $X_d^{PBL}$, which is defined as the average of the lower 10 percentile of its surrounding ±3 grid cells in the zonal direction and meridional direction ($7×7 = 49$ grid cells in total by taking the current grid cell as the center) considering the extensive variations of the orography in the Middle East. The daily regional background is built when more than 10 grid cells have valid retrievals in this domain. $A_d^{PBL}$ is the corresponding air density column in the PBL. The details to derive $X_d^{PBL}$ and $A_d^{PBL}$ can be found in Liu et al. (2021). The advantages of including $X_d^B$ are (1) it can be used to diagnose the

contribution of inhomogeneous background, especially over mountains and coastal
regions, and (2) the system biases between CAMS and TROPOMI, which leads to
biased $X_d^{PBL}$, is included in both and can be greatly reduced by subtracting $X_d^B$ from
$X_d^{PBL}$.
The daily wind field ($\vec{w}$) halfway the height of the PBL (PBLH) close to the overpass
time is obtained from the ECMWF. Wind speeds are constrained between 0 m/s to 10
m/s because the divergence method works when advective transport takes place, and
extremely high wind speed are unfavorable for a method based on the regional mass
balance. Local wind-field changes induced by complicated orography inevitably leads
to a certain pattern of wind divergence ($\overline{D_d^W}$), which further influence
$$D_d^S = \vec{w}\,\nabla(XCH_4^{PBL} - XCH_4^B) + (XCH_4^{PBL} - XCH_4^B)\,\nabla\vec{w} \quad (2)$$
Liu et al. (2021) corrected $E'$ by using an empirical correction by using a spatial
correlation between $\overline{D_d^S}$ and $\overline{D_d^B}$ to account for the effect of inhomogeneous background
and $\nabla\overline{w}$ over Texas, where the terrain is relatively flat and less affected by mountains.
To better reduce the effect of winds, we followed the method proposed by Sims (2018)
to iteratively remove the gradients of $\nabla\vec{w}$ on each day to get a non-divergent wind field,
$V$ component (south-north) and $U$ component (west-east), for the calculation of Eq. (1).
The positive values of $\overline{D_d^S}$ due to orography-raised wind near Tehran in Fig. 2d are
largely reduced (Fig. 2f) by using a non-divergent wind field. The magnitudes of $\overline{D_d^B}$ in
Fig. 2e also get close to $\overline{D_d^S}$. Before we applied this change, we tested the non-divergent
method in the GEOS-Chem simulation that was used in Liu et al., (2021). We found
that this step slightly improved the capability of the method in resolving the spatial
variability of sources (Figure S2), but underestimate the final emission by about 15%
in the GEOS-Chem simulation. In contrast, when deriving the emissions from
TROPOMI, using a non-divergent wind field especially improves the robustness over
coastal areas and typically increases emissions by 5-20% for most cases (Table S2
shows an example). The difference in change of emissions between GEOS-Chem
simulation and TROPOMI is primarily due to the correction of the final estimated
emissions. As was mentioned in the manuscript, the final emission based on the
divergence ($\overline{D_d^S}$). (Fig. 2d) apparently contains the residual of the divergence of
background ($\overline{D_d^B}$) (Fig. 2c), which is highly correlated with wind divergence ($\overline{D_d^W}$).
However, this dependence is much smaller for the GEOS-Chem simulation and for the
emissions derived from TROPOMI by using non-divergent wind. The procedure and
the evaluation of removing the wind divergence from the original wind field are
explained in Part B in SI. Generally, using a non-divergent wind field can improve the
capability of the method in resolving the sources, both in a model simulation and in
TROPOMI observations.
*2.3 Estimating emissions based on the divergence*
The inhomogeneous spatial distribution of $\overline{D_d^B}$ indicates the possible residue of the
regional background we built in Sect. 2.2. Therefore, we evaluate the contribution from
the residue background for each grid cell with positive $E'$ by checking the spatial
correlation between $\overline{D_d^B}$ and $\overline{D_d^S}$ in the domain that we defined to build the regional
background (its surrounding ±3 grid cell). For grid cells with positive $E'$, a linear
regression is applied to its surrounding ±3 cells:

$$y_i = k \cdot x_i + b \qquad (3)$$

where $y_i$ stands for $\overline{D_d^S}$ and $x_i$ stands for $\overline{D_d^B}$ of grid *i*. *k* and *b* are the slope and intercept
of the linear regression, respectively. If Eq. (3) is applicable to the center grid, it implies
the residue of the background still contributes to $E'$ and should be subtracted. This
linear correlation can be distinctive over locations with large variations in orography
(e.g., mountains, coastal areas). If more than 68% of the grid cells and the grid cell itself
fall within the prediction lines of Eq. (3), estimated emissions are set to zero because
$\overline{D_d^S}$ can be fully predicted by $\overline{D_d^B}$ according to Eq. (3). The grid cells are considered to
be influenced by residue background only when Eq. (3) is significant (p-value < 0.01),
and they are further corrected by the spatial correction:

$$E^{corr} = E' - (k \cdot \overline{D_d^B} + b) \qquad (4)$$

in which $(k \cdot \overline{D_d^B} + b)$ is regarded as the contribution from the remaining background,
which should be subtracted from the preliminary estimated emissions, $E'$. In addition,
we find that areas with negative $E'$ together with negative $\overline{D_d^B}$, implying no significant
sources exist. The final estimated emissions at grid cells with negative $E'$ are also set
to zero (Liu et al., 2021).

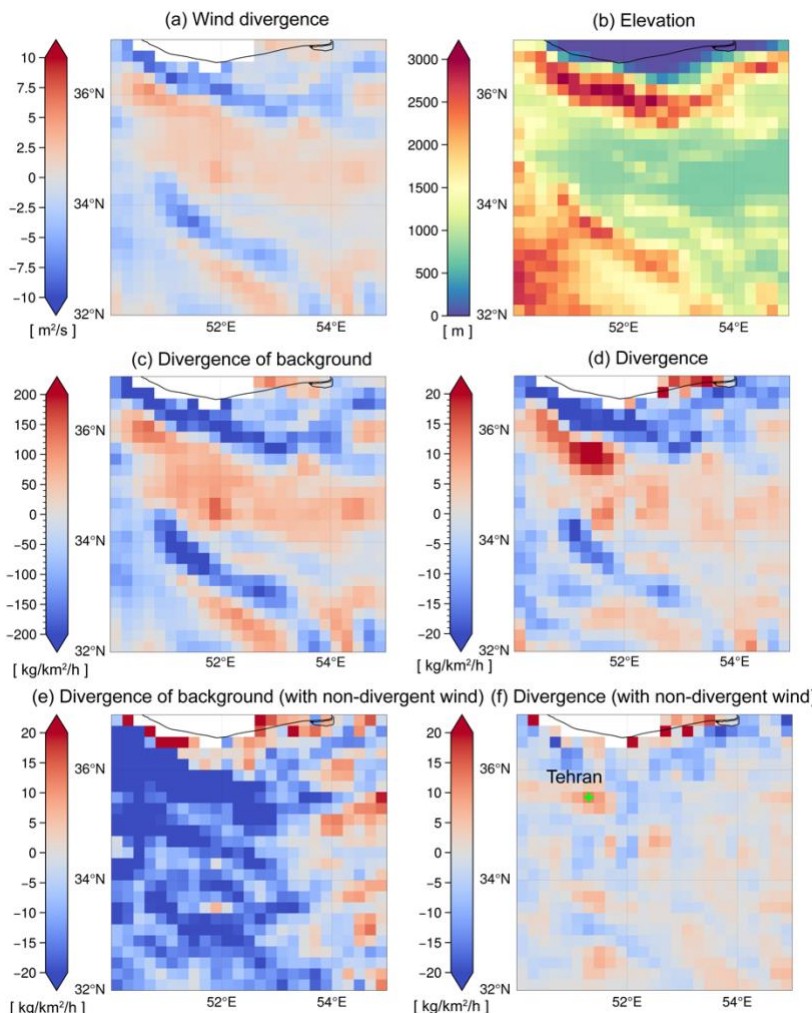

**Figure 2**. (a) The spatial distribution of original wind divergence ($\overline{D_d^W}$). (b) Elevation map generated from the GMTED2010 data set at 30 arcsecs (http://topotools.cr.usgs.gov/GMTED_viewer/). (c) Divergence of the background ($\overline{D_d^B}$) calculated with original daily wind field in 2019. (d) Divergence of methane enhancement ($\overline{D_d^B}$) under 500 meters with original daily wind field. (e)-(f) are similar to (c)-(d) but with the daily non-divergent wind field ($U$ and $V$). The green "+" in (f) is used to generate the time series of $D_d^B$ and $D_d^S$ in Figure 5b.


*2.4 Build temporal filter to identify possible false sources*

The artifacts caused by the variability of spectral albedo (e.g., specific soil types and interferences in the spectral range of the retrieval windows) have been generally

reduced in the WFMD v18 product (Schneising et al., 2023). The unrealistic
enhancements are reduced/removed over most locations. However, the biases
mentioned above can still exist in some places, as shown in Figure 3. In the northeast
near Riyadh, the stripe-shaped XCH4 enhancements (Fig. 3a) coincide with the
locations of high albedos (Fig. 3b) that cannot be explained by the changes of elevations
from southwest to northeast (Fig. 3c). The relevant correction has been done by
machine learning calibration in the WFMD v18 product, thus we found no universal
pattern that can be used to describe the relationship among XCH4, surface albedo and
aerosol. Therefore, we do not correct this kind of bias, following Liu et al. (2021), to
avoid double-correction. Alternatively, we try to find an objective way to filter false
emissions caused by retrieval artifacts.
A grid cell with a large $E'$ but no significant linear correlation between $\overline{D_d^S}$ and $\overline{D_d^B}$
contains either a source or is caused by artifacts in the retrieval, such as the case shown
in Fig. 3. If the enhancement is a kind of artifact; for example, caused by a bright surface,
it behaves more like a constant over days. Therefore, temporal variations of $D_d^S$ will be
mainly dominated by daily variations of the background, according to Eq (1).
Considering that the values of $D_d^B$ are much higher than $D_d^S$, as $XCH_4^{PBL}$ is used to
calculate $D_d^B$ while $(XCH_4^{PBL} - XCH_4^B)$ is used to calculate $D_d^S$, we normalize time
series of $D_d^S$ and $D_d^B$, respectively. This normalization allows for a better comparison of
their temporal variations (amplitudes). The temporal filter is based on their normalized
time series and built as follows. Firstly, we remove the grid cells that have less than 10-
day records. Next, if more than half of the days in the time series of a grid cell have a
normalized positive $D_d^S$ larger than $D_d^B$, the derived source (grid cell) is considered to
be real and not a retrieval artifact. . As an example, we take a grid cell (showing with a
green "+" in Fig. 3e) that is affected by the albedo near Riyadh. It has a larger $\overline{D_d^S}$ than
its surrounding grid cells, but the linear regression is not applicable here (p_value of
Eq. (3) is 0.2), suggesting the regional background we built is not biased. However,
only 20% (value of R in Fig. 4) of the total reliable days in 2019 have larger positive
normalized $D_d^S$ (Fig. 4b), indicating the daily variation is not significantly different
from its background. Hence the reliability of this source needs to be checked. In contrast,
more than 50% of the total days of the grid cell, which is verified as a true source in
Tehran (a green "+" in Fig. 3e), have larger positive normalized $D_d^S$. In this way, the
emissions from an artifact or random noise from the retrieval can be objectively
identified. In this study, we set the temporal filter such that at least more than 50%
observations from the time series have a larger positive normalized $D_d^S$ than the
normalized $D_d^B$.
However, we should also be aware that the threshold of the temporal filter used in this
study is relatively rigid, possibly excluding sources that occasionally release a large
amount of methane, like intermittent oil/gas leakage and inappropriately burned waste

gases. The preserved sources that pass the temporal filter are suggested to be more constant than that did not pass the temporal filter. For grid cells not affected by retrieval issues, the role of the temporal filter is more like an indication of the persistence or regional significance of a source, and the emissions without the temporal filter might, in some cases, be more realistic. The role of the temporal filter will be further discussed in Sect. 3

The divergence method requires sufficient temporal records (typically more than 7 days with valid observation for a grid cell) to derive robust results. Thus, the divergence on a single day does not provide a realistic emission for that day, and taking the standard deviations for individual days does not reflect the uncertainty or variability of a source. In addition, this method is not suitable for sources with a few intermittent releases, such as sudden leaks in oil and gas production. $\overline{D_d^S}$ can be a quite large positive value for this kind of source. However, a small number of large releases in a time series may lead to a removal of this source by the temporal filter (see the case of Fig. 6 in Sect. 4), which is built for automatically detecting retrieval artifacts over a large domain. In order to keep as many real sources as possible, we apply a Monte Carlo experiment to each possible source to estimate the uncertainty of the derived emissions and to evaluate the robustness/reliability of a source. The procedure is as follows:

(1) We randomly choose 80% of the sampling days from a time series in a year as a subset. We derive a new emission, $E_i$, and count the ratio, $R_i$, of the number of days that have larger normalized $D_d^S$ than normalized $D_d^B$.

(2) Repeat step (1) 30 times for a time series that has more than 20 sampling days while 10 times for the one that have fewer days to derive the set of emissions, $\{E_i\}$, and the set of ratios, $\{R_i\}$ for each possible source. $R_i$ is used as the temporal filter in each subset.

(3) Take one-standard deviation of the set $\{E_i\}$ as an uncertainty of a source. If the median value $(R)$ of $\{R_i\}$ is greater than 0.5, this source is regarded having high confidence, which means these emissions are constantly released and likely not caused by a retrieval artifact.

We also investigate the choice of the percentage of the time series and the number of the iterations. 80-70% percent can be a reasonable range that ensure the representativeness as well as randomness of sampling days. We have tested the number of iterations from 10 to 50 times. The uncertainty map such as Fig. 5c become stable

after 20 iterations, and 30 iterations can ensure the robustness as well as the efficiency
of the calculation.

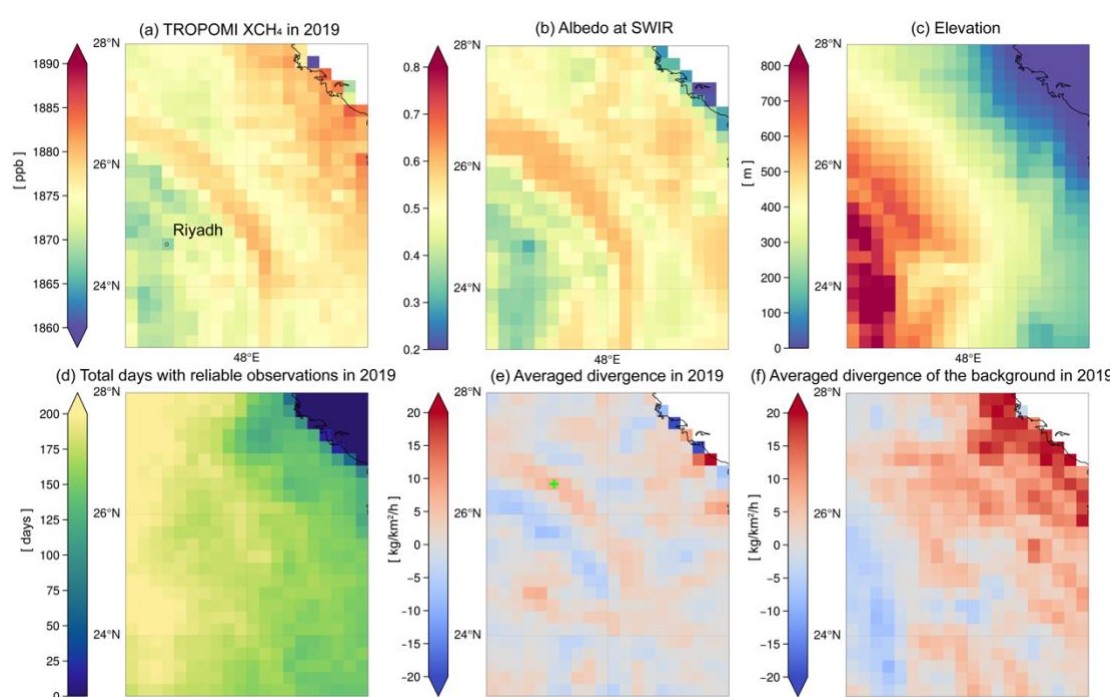

**Figure 3.** Gridded 0.2° × 0.2° annual average of (a) TROPOMI observed XCH₄ and
corresponding (b) TROPOMI apparent albedo at the short-wave infrared wavelength
(SWIR). (c) The gridded elevation map that is generated from the GMTED2010 data
set at 30 arcsec (http://topotools.cr.usgs.gov/GMTED_viewer/). (d) The total number
of valid observation days in 2019. (e) Averaged daily divergence ($\overline{D_d^S}$) and (f)
divergence of the background ($\overline{D_d^B}$) in 2019. The green "+" in (e) is used to generate
the time series of $D_d^B$ and $D_d^S$ in Figure 4(a).

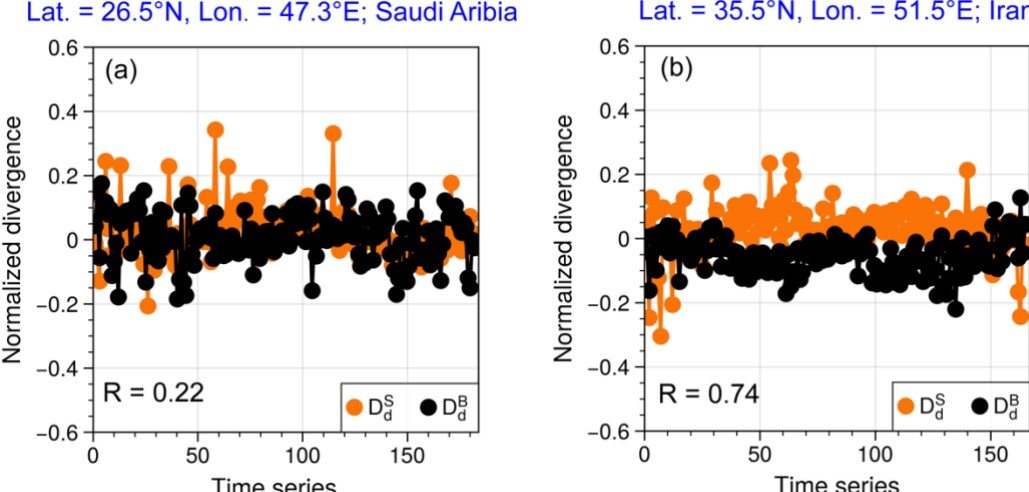

**Figure 4.** The time series of normalized $D_d^S$ (orange line) and $D_d^B$ (black line) of the grid cell in (a) Saudi Aribia and (b) Iran. The "R" in the lower left corner stands for the ratio of the number of days with a larger positive normalized $\overline{D_d^S}$ than $\overline{D_d^B}$ related to the total number of sampled days.

## 3 Results

*3.1 Deriving the final emissions with the temporal filter*

After we derived emissions based on the divergence, the possible false sources are further identified by the temporal filter. The strict temporal filter is introduced to objectively exclude artifacts related to retrieval issues. However, to a grid cell that is not affected by retrieval issues, the temporal filter acts more like an indication of the persistence of a source. Namely, methane is intermittently released from this source. Here we selected two areas in the Middle East to illustrate the role of the temporal filter in the emission estimation. Our methane annual emissions are then compared with three widely-used methane emission inventories in the same year, 2019. Other auxiliary datasets such as $NO_X$ emission inventories, methane plume complexes detected by EMIT imaging spectrometer and heating sources identified by VIIRS are also used to better evaluate our derived emissions.

Figure 5a and c show all possible sources and their relative uncertainties, respectively. Fig. 5b shows the final emissions after excluding the grid cells with emissions less than 3 kg/km$^2$/h, which is used as detection threshold of a source in this study. It is estimated by using the detection threshold of TROPOMI XCH$_4$ (Hu et al., 2018, Schneising et al., 2023) and the approach in Jacob et al., (2022). The detection threshold of methane source from TROPOMI is depending on many factors such as source types, inversion

methods and temporal coverage over a location etc., which can vary from ~0.5 kg/km$^2$/h
to 12.5 kg/km$^2$/h (Lauvaux et al., 2022; Dubey et al., 2023; Jacob et al., 2016; 2022).
Fig. 5a suggests presence of small sources around the center of Riyadh, where a number
of heating sources are detected by VIIRS. Additionally, small sources are detected in
the south to Riyadh, where dairy farms and industry areas are located. The spatial
distributions over two areas are similar to the DECSO NO$_X$ emissions, indicating
existence of human activities. However, we found that sources below the detection
threshold show large uncertainties (>20%) in this study, which means the method is not
robust to distinguish these small sources from the regional background.
Both constant sources and artifacts (the "stripe" in the north of Riyadh) show small
relative uncertainties (Fig.5c) due to continuous regional enhancement of XCH$_4$. Only
a few sources pass the temporal filter in the middle of Saudi Arabia (marked by blue
"+" in Fig. 5b, indicating they are with high confidence). However, some facilities are
found over the Khurais oil field in Google Earth image while it fails to pass the temporal,
indicating they might be true but not constant. Another similar case is in the middle of
the Syria Arab Republic, where many methane plumes along the Euphrates River are
detected by the EMIT instrument (Fig. 6b) but reported quite low by three bottom-up
emission inventories. They are reported as non-continuous sources (fail to pass the
temporal filter) in our emission inventory (Fig. 6a). Thus, applying the strict temporal
filter in an area without retrieval issues is aim at identifying continuous sources. In
addition, except for the capital, Riyadh, both EDGAR and CEDS show that the primary
type of sources in Saudi Arabia is energy related. The locations of oil/gas-related fires
also match well with the sources of methane in the eastern area in Fig. 5g. However,
our estimates (Fig. 5b) and methane emissions from the fuel exploitation reported by
GFEI v2 (Fig. 5f) are quite low (lower than the TROPOMI detection threshold) in the
eastern oil/gas production area. This finding is similar to the result of Lauvaux et al.
(2022) that fewer ultra-emitters of methane are detected by using the TROPOMI CH$_4$
operational product (Lorente et al., 2021) in Middle Eastern countries such as Kuwait
and Saudi Arabia, which could be attributed to fewer accidental releases and/or
stringent maintenance operations. Using the locations and frequency of flares to
estimate the methane emission in bottom-up emission inventories could have led to
overestimation of the methane emissions in this region.
In contrast, Figure 7 show the case over Tehran and its surroundings. Most sources in
this area pass the strict temporal filter, indicating they are quite constant. Five areas are
identified as hotspots of methane sources in Fig. 7b. Fig. 7d-f shows the spatial
distributions of methane sources estimated by EDGAR, CEDS and GFEI in 2019. The
bottom-up emission inventories show lower methane emissions than our results. The
dominant category of methane sources in this area is not energy-related but others like
waste treatment and agriculture (see classification in Table-1), as suggested by EDGAR
and CEDS. A number of heat sources due to metal or non-metal industry production are
also identified by VIIRS over these hotspots. A good match in locations between
methane and NO$_X$ sources over Tehran, Isfahan, and Atarabad is found when we further
examine NO$_X$ source distributions in EDGAR and DECSO. One possible reason for the
consistence over these areas can be that the methane emissions may come from waste
treatment in cities, where landfilling is the most common way of municipal solid waste
(MSW) disposal in Iran (Pazoki et al., 2015). Fig. 7c presents a case of methane plume
identified by EMIT instrument on 23$^{th}$ April 2023 near Kashan power plant that is
apparently not reported in current inventories. Actually, some facilities have been found
in Google Earth images near Kashan, which are also identified by our method in Fig.
7b. Another hotspot area located between Tehran and Kashan is near Kavir National
Park, where we currently have no clear explanation for the emissions.

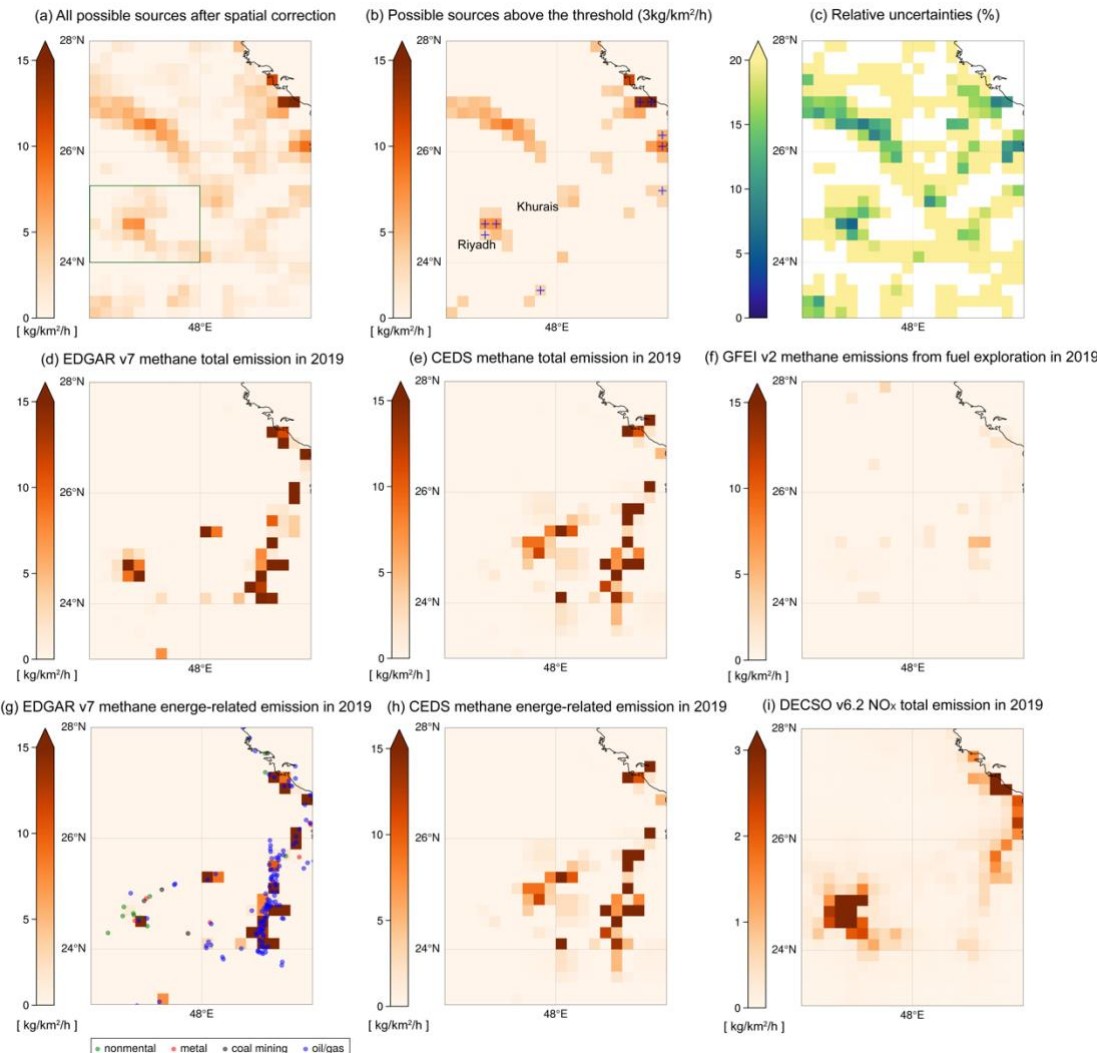

**Figure 5**. (a) Averaged annual methane emissions derived from the divergence after the spatial correction in the middle of Saudi Aribia. (b) All possible sources above the detection threshold of emissions in this study ($3kg/km^2/h$). Grid cells that pass the temporal filter are marked by blue "+". (c) The relative uncertainty of derived methane emissions in (a). (d) EDGAR v7.0 averaged annual methane total emission in 2019. (e) CEDS v_2021_04_21 averaged annual total methane emissions in 2019. (f) GEFI v2 averaged annual methane emissions from fuel exploration in 2019. (g) Energy-related methane emissions from EDGAR v7.0 overlapped with the industrial heat sources identified by VIIRS instrument. (h) CEDS v_2021_04_21 energy-related methane emissions in 2019. (i) Averaged annual DECSO v6.2 NO$_X$ total emission in 2019. The spatial resolution of all emission data showing here is $0.2° \times 0.2°$.

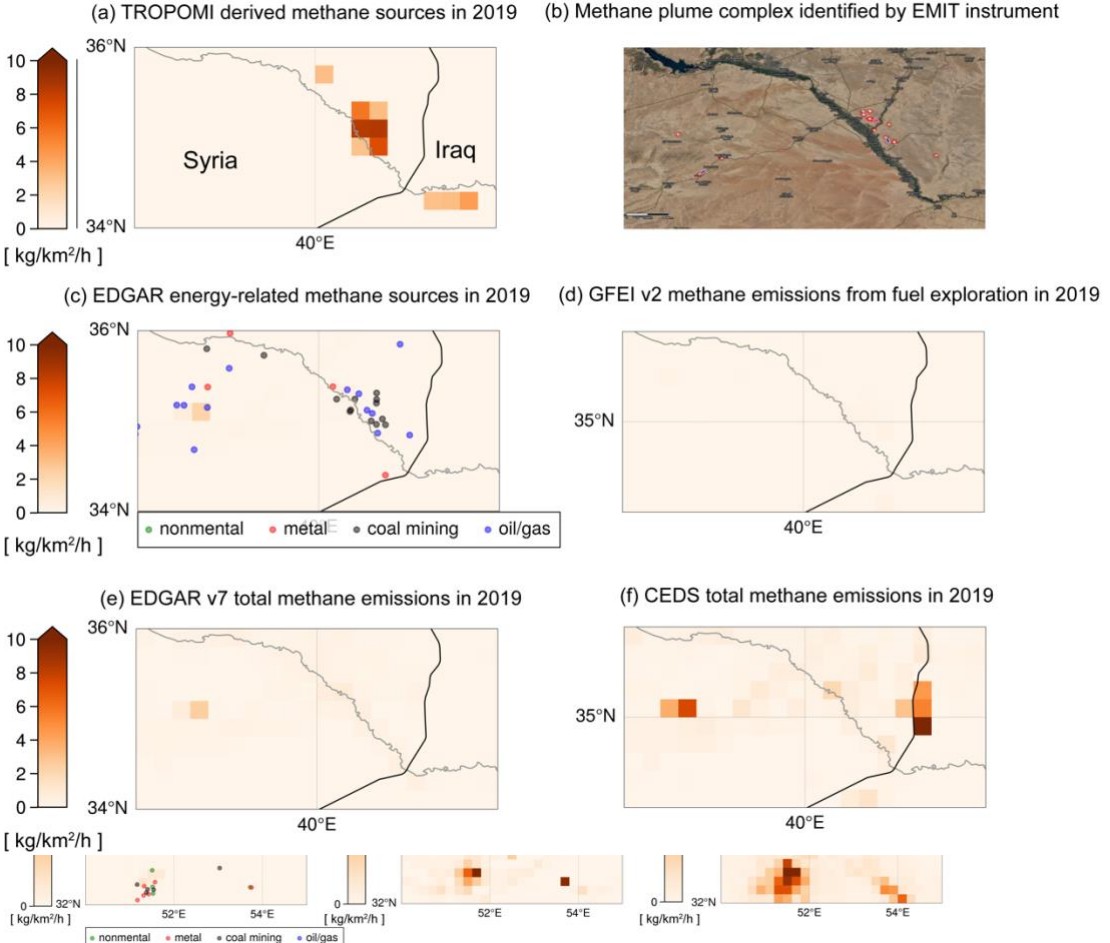

**Figure 6**. (a) Averaged annual methane emissions over Syria from TROPOMI
observations in 2019. (b) The detected methane plume complex (red circles) by the
EMIT instrument. (c) Energy-related methane emissions from EDGAR v7.0
overlapped with the industrial heat sources identified by the VIIRS instrument. (d)
GEFI v2 methane emissions from the fuel exploitation in 2019. (e) EDGAR v7.0
emission inventory in 2019. (f) CEDS v_2021_04_21 total methane emissions in 2019.
The spatial resolution of all emission data showing here is 0.2° × 0.2°.
**Figure 7**. (a) The spatial distribution of TROPOMI observed $XCH_4$ in 2019 on a grid
of 0.2°. (b) The methane sources derived from TROPOMI after the spatial correction
and are higher than 3kg/km$^2$/h (inferred from the detection threshold of TROPOMI
$XCH_4$). The grid cells with high confidence, passing the temporal filter, are marked by
a blue "+". (c) The detected methane plume complex by the EMIT instrument in Kashan
on 23th April 2023 (Source: https://earth.jpl.nasa.gov/emit-mmgis-lb/?s=e7z1z). (d)
EDGAR v7.0 averaged annual methane total emission in 2019. (e) CEDS
v_2021_04_21 averaged annual total methane emissions in 2019. (f) GEFI v2 averaged
annual methane emissions from the fuel exploitation in 2019. (g) Energy-related
methane emissions from EDGAR v7.0 overlapped with the industrial heat sources
identified by the VIIRS instrument. (h) Averaged annual EDGAR v6.1 NO$_X$ total
emission in 2019. (i) Averaged annual DECSO v6.2 NO$_X$ total emission in 2019.

 *3.2 Annual CH₄ emissions over the Middle East based on TROPOMI*

In Figure 8, we select five hotspot regions in the Middle East to further assess the annual
regional emissions from 2019 to 2022. Before we calculate the emissions of each region,
we checked spatial patterns of $XCH_4$ and albedo from TROPOMI, as well as land
features, to ensure no suspicious retrieval artifact is included as a source. The emissions
are based on all possible sources and only confident sources are shown. The results of
all possible sources (pink bars) may be more representative of the total emissions in
these areas, and the emissions passing the temporal filters (blue bars) can be used to
estimate the contribution of constant sources. Here we should clarify that the constant
source in our paper does not refer to one with a constant emission factor but indicates
a source that continually releases methane for most days of a year. The areas used to
calculate annual emissions (bars in Fig. 8) are shown as dark green rectangles in the
insets on the top. The emission map in each panel of Fig. 8 is the annual methane
emissions of EDGAR v7.0 in 2019. The energy-related sectors and the other categories
(waste, agriculture, and transportation) of EDGAR v7.0 methane emissions from 2018
to 2021 are displayed by the first stacked green/yellow bars in Fig. 8a–e. The category-
based annual emissions of CEDS in 2018 and 2019 are shown in the last stacked
purple/orange bars. The estimate of GFEI for the fuel exploration in 2019 is shown as
a red asterisk overlapped on the third column. We should clarify that our estimate for
the total emission in each year is the sum of sources that are higher than $3kg/km^2/h$ in
the study area, but the total emission reported by a bottom-up emission inventory
includes grid cells with emissions across all ranges. Thus, theoretically our estimates
will underestimate the real emissions.
The main type of methane sources in Tehran and Isfahan given by EDGAR and CEDS
is waste, and the energy-related sources are not oil/gas production based on VIIRS
detected fire types and EDGAR's prediction (Fig. 7g). The derived methane emissions
are also more constant. Smaller differences are found between the blue and pink bars
than Riyadh, West of Turkmenistan and Iran & Iraq (Fig. 8c-e). Our estimates in Tehran
are 12-30% higher and 33-52% higher than EDGAR's and CEDS's estimates for
constant sources, respectively. Our result (220 kt/yr for 2018-2021) is much lower than
the emission estimated by de Foy et al., (2023) (953 kt/yr for 2017-2021) over Tehran,
which is 8.3 times higher than EDGAR v6.0's estimates (114 kt/yr) used in that paper.
The possible reasons could be different assumptions of the regional background and the
methods to calculate the emission of the area. The Gaussian model used by de Foy et
al., (2023) treated an urban area as one large source and integrated the emissions along
the "plume", whereas our total emission for a certain area is the sum of individual
sources that are derived from the divergence method. GEFI's estimate for the fuel
exploration is 2-3 times higher than EDGAR's and CEDS's estimates, indicating
possible underestimations of the two inventories in Tehran. The sources in Isfahan,
another Iranian metropolis, are also constant over time (very small difference between

blue and pink bars). However, our derived emissions are about 3 times higher than the two inventories. Sources in our inventory are distributed over a wider area in Isfahan, and their spatial distributions are similar to $NO_X$ sources of EDGAR and DECSO, indicating the emissions are very likely from activities in the city. Although Isfahan has been attempting to gradually transform the landfill-based disposal system into a modern system with less production of greenhouse gases, the high methane emissions we derived might also imply that waste management is still a challenge (Abdoli et al., 2016). A similar result was found by Chen et al. (2023), in which they found waste emissions could be underestimated by more than 50% in certain Middle Eastern countries like Iran, Iraq, and Saudi Arabia.

The total constant emissions we derived for Riyadh are half that of EDGAR but close to CEDS's estimate. As shown in Fig. 5, the spatial distributions of various inventories can be very different. The domain we used to calculate the total emission is defined by the spatial distribution of EDGAR, but oil/gas-related flares are located in the northeast of Riyadh (blue dots in Fig. 5g). However, including these cells only increases total emissions by 5–8% because they are smaller than $3kg/km^2/h$ therefore below the detection threshold of TROPOMI. Moreover, ~50% of the emissions in Riyadh are constant (have constant emission factor), which can be another reason of the large discrepancy between different inventories.

Western Turkmenistan near the Caspian Sea and the coastal regions of Iran and Iraq are two well-known oil/gas production areas in the Middle East. The energy-related sectors (green bars) contribute more than 92% in the two regions based on EDGAR estimates. The constant emissions derived from TROPOMI (blue bars) in the west of Turkmenistan are quite comparable to GFEI's estimate but nearly two times higher than estimates of EDGAR and CEDS. Although total methane emissions estimated by EDGAR and CEDS are very similar, the spatial distributions of sources are different (Figure S3). The constant sources of oil/gas there contribute to ~55% of the total emissions over the four years based on our estimates, which agrees with Varon et al. (2021), who concluded the sources here are intermittent, and the persistence rate is ~40%. Our estimates will be four times higher than the total emissions of these two inventories if all possible sources are included. The large uncertainty also implies that resolving the sources here can be quite difficult because of the few observations near the coast and the variabilities of the sources.

The annual variations in the coastal area of Iraq and Iran are consistent in EDGAR's and our estimates (the offshore emissions in bottom-up emission inventories are ignored because the observation of TROPOMI over ocean can be quite difficult). It increased to surpass the total emission of 2018 in 2021 after a modest decline from 2018 to 2020. The fraction of constant sources is much less than in Western Turkmenistan. Our estimates are comparable to EDGAR if all possible sources are included. However, the total emissions from constant sources are quite low, and they

are comparable to the other methane emissions estimated by CEDS, which mainly come
from waste and are quite low in EDGAR estimates. Chen et al. (2023) found that oil/gas
emission derived from their inverse modeling with the TROPOMI observation is 43%
and 58% lower than in their bottom-up emission inventory over Iran and Iraq,
respectively. Lauvaux et al. (2022) also showed fewer ultra-emitters of methane are
detected by using the TROPOMI $CH_4$ operational product (Lorente et al., 2021) in
Middle Eastern countries such as Kuwait and Saudi Arabia, which could be attributed
to fewer accidental releases and/or stringent maintenance operations. Thus, for an area
with many occasionally released methane, using a constant emission factor or flaring
data as an index may lead to an overestimation of methane leakage from the oil/gas
industry. In addition, we checked plume complexes detected by EMIT, and find that the
max value of each plume complex can differ by an order of magnitude, implying the
large variabilities of released methane here. The coarse spatial resolution of our
emission data may smooth plume complexes and can be another reason of predicted
lower emissions.

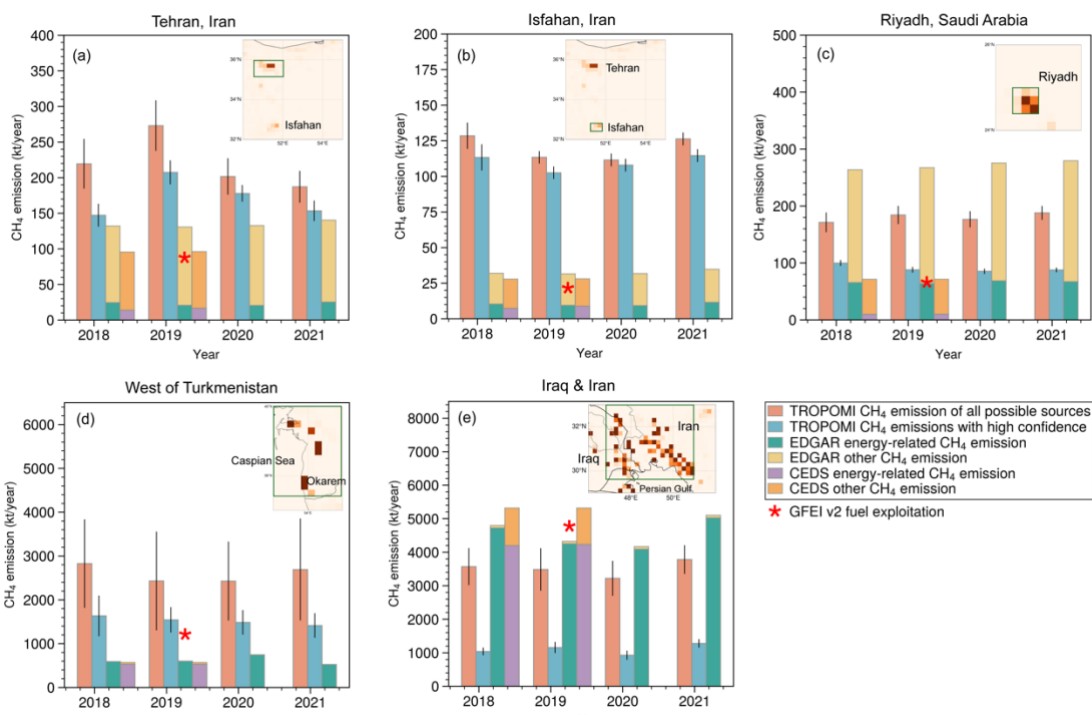

**Figure 8**. Regional total methane annual emissions estimated by EDGAR v7.0 and
TROPOMI from 2018 to 2021. The areas used to generate bars in (a–e) are shown in
dark green rectangles in embraced emission maps of total emissions of EDGAR in 2019.
The ranges in latitudes and longitudes can be found in Table S1 in SI. A green bar
represents the energy-related emissions, and a yellow bar represents the remaining
methane emissions in EDGAR v7.0. A purple bar represents the energy-related
emissions, and an orange bar represents the remaining methane emissions in CEDS
v_2021_04_21. The blue bar is the total emission of sources that pass the temporal filter
and are higher than 3kg/km$^2$/h. The pink bar represents the total emission of all possible

sources that are higher than 3kg/km$^2$/h. All the emissions over water (the Caspian Sea
and the Persian Gulf) are ignored because of too few observations and large
uncertainties. An error bar represents the sum of uncertainties associated with each
source in this area. The calculation of the uncertainty of a source (grid cell) is presented
in Sect. 2.4.

**4 Conclusions**

An improved divergence method using non-divergent wind fields with a temporal filter
has been developed to better estimate CH$_4$ emissions from observations of the
TROPOMI instrument over areas with complicated orography and/or high albedo, like
the Middle East. The non-divergent wind largely reduces the biases caused by drastic
topography changes. The residue of the background (e.g., sources in Tehran, located in
a valley) is further subtracted from the emission through spatial correction. The
temporal filter is built to further exclude false sources due to retrieval issues. It also can
be used to test the persistency of sources over an area free of artifacts. We found that
emissions from wastes (e.g., landfills, wastewater) or agriculture (e.g., livestock farms)
can be quite persistent in time compared to the oil/gas-related sources in the Middle
East.
We further compared our annual regional total emissions with EDGAR v7.0, CEDS
v2021_04_21 and GFEI v2 for various regions in the Middle East with different source
categories from 2018 to 2021. The oil/gas productions at the coast of Iran and Iraq are
quite intermittent compared to the west of Turkmenistan where our estimate for
constant sources is quite comparable to the emission from the fuel exploitation
estimated by GFEI v2. The continuous release of methane from waste or farms can
contribute considerably to the total methane emissions in several metropolises in the
Middle East, which can be two times higher than EDGAR's and CEDS's estimates.
In future work, the role of the temporal filter can be largely reduced with new improved
retrieval products of TROPOMI CH$_4$. This will especially allow better estimates of
intermittent methane emissions.
*Acknowledgments*
*Competing interests.*
The authors declare that they have no competing interests.
*Funding.*
ESA project IMPALA, grant number: 4000139771/22/I-DT-bgh
*Author contributions.*
ML, RVA, and MVW designed the experiment and analyze the results. ML performed
all calculations and visualized the results. The codes for estimating methane emissions
are mainly developed by ML and are supported by LB, HE and PV. HK and JD help to
visualize the results. The wind fields are extracted by HE. YL provides the category-
related VIIRS data. All co-authors contributed to review the manuscript.
*Data and materials availability:*
TROPOMI/WFMD v1.8 methane Level-2 dataset is available at:  https://www.iup.uni-
bremen.de/carbon_ghg/products/tropomi_wfmd/
EAC4 of CAMS, which used to be estimated the column above the PBL can be accessed
at:      https://ads.atmosphere.copernicus.eu/cdsapp#!/dataset/cams-global-reanalysis-
eac4?tab=overview
EDGAR v7.0 for methane anthropogenic emissions and EDGAR v6.1 for $NO_x$
anthropogenic            emissions            are            available            at:
https://edgar.jrc.ec.europa.eu/overview.php?v=432_GHG
CEDS  v_2021_04_21  for  methane  anthropogenic  emissions  is  available  at:
https://data.pnnl.gov/dataset/CEDS-4-21-21
GFEI  v2  for  the  methane  emissions  from  fuel  exploitation  is  available  at:
https://dataverse.harvard.edu/dataset.xhtml?persistentId=doi:10.7910/DVN/HH4EUM
&version=2.0
MODIS daily 10km AOD data can be downloaded through NASA Earthdata portal:
https://search.earthdata.nasa.gov/search
DECSO total anthropogenic $NO_x$ emission is available at: www.globemission.eu
The  CH4  plume  complexes  detected  by  EMIT  instrument  are  available  at:
https://earth.jpl.nasa.gov/emit/data/data-portal/Greenhouse-Gases/

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
