# Peer review of "Current potential of CH4 emission estimates using TROPOMI in the Middle East"

_EGUsphere, 2024_

## Author Comment (AC1)

We appreciate the feedback from the reviewer. The comments from the reviewer are displayed in blue text, along with our responses, which are displayed in black text.

**Response to RC1:**

This paper uses an improved flux divergence algorithm to estimate sources of CH4 in the middle east. The results are compared with different emission inventories, and discrepancies are noted with both over and under estimations of emissions. Overall, the paper is well written and informative, the method seems sound and the results are interesting, timely and relevant. I am happy to recommend publication.

**General comments**

One weakness was some missing citations of recent work that is relevant. You cite Beirle et al., 2019, but should really also consider Beirle et al., 2022 which contains version 2 of the flux divergence method. Sun, 2022, provide further developments to the method. Although both of these are for NO2, they are for the same sensor and there is sufficient overlap with CH4 that merits their mention. De Foy and Schauer, 2023 estimate CH4 emissions in urban areas, including Tehran. It would be interesting to see the difference in estimates. Finally, Roberts et al., 2023 specifically look into the impact of missing data on CH4 flux retrievals. There may be further papers – it would be good to compare this paper with the latest publications.

We thank the reviewer for providing us the relevant information. Beirle et al., (2023) and Sun et al., (2022) have been added to the revised manuscript at L68. We believe the conclusion about the impact of missing data on the divergence method in Roberts et al., (2023) is not very suitable to be cited in our study because it specifically examines the impact of missing data at a very high spatial resolution (the typical radius of an emission field is 200-400 meter).

The emission estimated by de Foy et al., (2023) for 2017-2021 is 953 kt/yr over Tehran based on Table C1, which is 8.3 times higher than EDGAR v6.0's estimates (114 kt/yr) used in that paper. Our estimate over Tehran is 220 kt/yr for 2018-2021, which is about 2 times higher than EDGAR v6.0 and much closer to the conclusion in that paper: "We show that methane emissions from urban areas may be underestimated by a factor of 3–4 in the EDGAR greenhouse gas emission inventory". The possible reasons could be different methods used to determine the background and calculate the total emission of the area. The Gaussian model in that paper treated the urban area as one large source and integrated the emissions along the "plume", whereas our total emission for a certain area is the sum of individual sources derived from the divergence. We added this comparison in our revised manuscript at L515:

"Our result (220 kt/yr for 2018-2021) is much lower than the emission estimated by de Foy et al., (2023) (953 kt/yr for 2017-2021) over Tehran, which is 8.3 times higher than

EDGAR v6.0's estimates (114 kt/yr) used in that paper. The possible reasons could be different assumptions of the regional background and the methods to calculate the emission of the area. The Gaussian model used by de Foy et al., (2023) treated an urban area as one large source and integrated the emissions along the "plume", whereas our total emission for a certain area is the sum of individual sources that are derived from the divergence method."

Line 167 to 187: I was a bit skeptical of the boundary layer treatment: fixing PBLH at 500m seems rather crude. However, I notice that you published this already in your previous paper. I do wonder what would happen if you used actual PBLH from ERA5,

Liu et al., (2021) found that enhancements of $XCH_4$ due to the transport in the upper atmosphere should be removed before calculating the divergence, as they are irrelevant to the ground-level sources. Therefore, identifying the height of PBL is not very important, but estimating the $XCH_4$ in lower atmosphere is more relevant to the ground emissions. We used surface pressure, $XCH_4$ and total dry air density from TROPOMI observation, as well as the $XCH_4$, temperature and relative humidity profile from EAC4 (60 layers for the whole atmosphere, whereas TROPOMI uses 12-layer profiles for retrieval, which is too coarse to resolve meteorological dynamics), to estimate $XCH_4$ below 500 meters (referred as "PBLH" in our paper). The favorable height is suggested to be 500-700 meters above the ground considering the systematic difference between EAC4 dataset and TROPOMI observations (Liu et al., 2021). Using either a too shallow or too thick layer as "PBLH" can magnify the bias. To reduce the uncertainties caused by singular values in model simulation, we fixed the height at 500 meters. The way to describe the "500 meters" as "PBLH" in our original manuscript is indeed a little confusing. We addressed this in the revised paper at L191 and L201:

"Estimating the $XCH_4$ in lower atmosphere is quite important since the enhancement due to the transport in the upper atmosphere is irrelevant to the ground emissions."

"We fixed the PBLH at 500 meters above the ground considering the PBLH from the reanalysis dataset has large uncertainties and is occasionally too shallow (Guo et al., 2021). The favorable height is suggested to be 500-700 meters above the ground considering the systematic difference between EAC4 dataset and TROPOMI observations (Liu et al., 2021)."

Coastal regions present a particular challenge in terms of data filtering. Some of your figures do seem to suggest that there are anomalous retrievals near coastlines. It might be that more careful filtering of boundary retrievals is necessary compared with the default land/water mask used. This could be discussed for future reference.

Actually, we applied a new mask to identify water, land and the boundary, but that was not mentioned in the main text. Here we added the content in our revised manuscript at L116:

"Another aspect that is addressed is the distinction between land and water bodies, especially over the coastlines. TROPOMI use different retrieval strategies for data over land and ocean. The retrievals over ocean are only available in sun glint mode. We find the data over ocean can be quite noisy. Furthermore, the data continuous from land to ocean are checked. We selected pixels locating at several 1° ×1° areas covering half land and half ocean at the coastlines of Oman, Yemen and along the Red Sea. We found there are not many differences between pixels over land and ocean (see Figure S1 in SI). Therefore, we built a water-land mask at the same spatial resolution as our emission data (0.2° ×0.2°) based on Global Land Cover Characterization (GLCC) of the United States Geological Survey (USGS) (United States Geological Survey, 2018a, b) to distinguish water, land and the coast (transition grids from land to water). Only grid cells that are marked as land and coast are used to build the regional background and are used to calculate the daily divergence."

Line 325: Is it really feasible that a farm in the desert would produce a detectable amount of CH4? I think that this should be backed up with a bit more information if it is to stay here – information about the agricultural emissions, comparisons with farms with known CH4 emissions, threshold values for TROPOMI.

Actually, there is not only one farm in the grid cell. At least, there are about 135.000 cattle in six big farms in this area through Google Earth Image (see the figure below), one of them is the AlMarai farm which is one of the biggest dairy farms in the world (Shadbolt et al., 2013). But indeed, we do not have observation or reference to directly compare with.

[Figure]

Reference:
Shadbolt, P. : Milking the desert: How mega-dairies thrive in Saudi sands, CNN news, available at: https://www.cnn.com/2013/12/18/world/meast/milking-the-desert-saudi-dairy-farms (Last access: 16 June, 2024), December 18, 2013

**Minor points:**

Fig. 5g&h: "Mehttane"

Changed.

Line 137: despite *the fact* that the three

Changed

Line 449: 3 kg/m2 (remove extra /)

 Changed to "3kg/km$^2$/h"

**Suggested References:**

Beirle, S., Borger, C., Jost, A. and Wagner, T., 2023. Improved catalog of NOx point source emissions (version 2). *Earth System Science Data Discussions*, *2023*, pp.1-37.

de Foy, B., Schauer, J.J., Lorente, A. and Borsdorff, T., 2023. Investigating high methane emissions from urban areas detected by TROPOMI and their association with untreated wastewater. *Environmental Research Letters*, *18*(4), p.044004.

Sun, K., 2022. Derivation of Emissions From Satellite-Observed Column Amounts and Its Application to TROPOMI NO2 and CO Observations. *Geophysical Research Letters*, *49*(23), p.e2022GL101102.

Roberts, C., IJzermans, R., Randell, D., Jones, M., Jonathan, P., Mandel, K., Hirst, B. and Shorttle, O., 2023. Avoiding methane emission rate underestimates when using the divergence method. *Environmental Research Letters*, *18*(11), p.114033.

---

## Author Comment (AC2)

We appreciate the feedback from the reviewer. The comments from the reviewer are displayed in blue text, along with our responses, which are displayed in black text.

**Response to RC2:**

This study presents an interesting attempt to further improve the divergence method for estimating methane fluxes, by correcting for divergence in the dynamical flow and temporal filtering of retrieval artefacts. The divergence concept is interesting, but can be puzzling in its implementation also. It is important that this is done well, and doesn't overlook anything important (such as the assumption of stationary state, which is probably satisfied reasonably well but still a simplifying assumption). Promising results are obtained suggesting that the method works, and yields useful additional information about the intermittency of emissions. However, as explained further below, the proof that emissions that look better are indeed better is missing. This makes the validation of the proposed improvements currently too weak in my judgment. Besides this most important point of my review there are a few other issues to clarify, including the derivations of estimation uncertainties. With those issues solved I do not see a reason to uphold publication, but it is important that it is carefully done.

**General comments**

In Liu 2021 a great job was done validating the divergence implementation to methane using GeosChem. Those results looked promising, but also suggested room for improvement. It would be interesting to know if the improvements that are proposed here improve the comparison presented there (which has the same issues with elevation, surface albedo influences could easily be mimicked). This raises the question why it was not done. This concerns not only the estimation of emissions, but also the corresponding emission uncertainties. It is not obvious to me that altering the wind field to make it divergence free improves the comparison between this simplified 'model' of the atmosphere and the TROPOMI observations, unless the observations themselves are corrected for the influence of dynamical divergence influences.

We thank the reviewer for their comment. Actually, we did test the non-divergent wind in the GEOS-Chem simulation we conducted in Liu et al. (2021), but it was not mentioned in the original manuscript.

As the reviewer expected, using non-divergent winds did not significantly change the ability of the divergence method to identify sources. And our result shows that the non-divergent method slight improved the capability of the method in resolving the spatial variability of sources (see results of Reduced Major Axis regression (RMA) in Fig. S2(e) and (f), also attached below), but slightly underestimate the quantity of sources (slopes) in the "ideal case". Additionally, we analyzed emissions from TROPOMI in the Middle East to compare the effects of non-divergent wind presence and absence. Using a non-divergent wind field especially improves the robustness over coastal areas and typically

increases emissions by 5-20% for most cases. We also added the results over five hotspot areas in Table S2 as an example.

**Table S2. The annual emissions derived with and without non-divergent wind**

| West of Turkmenistan [37.0°N, 53.0°E, 40°N, 55°E] | | | | |
|---|---|---|---|---|
| kt/yr | 2018 | 2019 | 2020 | 2021 |
| EDGAR total | 592 | 601 | 746 | 525 |
| All sources (non-divergent $w$) | 2826 | 2429 | 2426 | 2692 |
| All sources (uncorrected $w$) | 691 | 676 | 663 | 540 |
| Tehran [35.2°N, 50.6°E, 36°N, 52°E] | | | | |
| kt/yr | 2018 | 2019 | 2020 | 2021 |
| EDGAR total | 132 | 131 | 133 | 140 |
| All sources (non-divergent $w$) | 219 | 273 | 202 | 187 |
| All sources (uncorrected $w$) | 202 | 214 | 165 | 185 |
| Isfahan [32.4°N, 51.2°E, 32.8°N, 52.0°E] | | | | |
| kt/yr | 2018 | 2019 | 2020 | 2021 |
| EDGAR total | 32.00 | 32 | 32 | 35 |
| All sources (non-divergent $w$) | 129 | 113 | 112 | 126 |
| All sources (uncorrected $w$) | 122 | 87 | 104 | 113 |
| Iraq & Iran coastal area [29.6°N, 47.0°E, 32.6°N, 51°E] | | | | |
| kt/yr | 2018 | 2019 | 2020 | 2021 |
| EDGAR total | 4796 | 4327 | 4168 | 5102 |
| All sources (non-divergent $w$) | 3570 | 3484 | 3220 | 3781 |
| All sources (uncorrected $w$) | 3079 | 2886 | 2363 | 3213 |
| Riyadh [24.4°N, 46.4°E, 25°N, 47°E] | | | | |
| kt/yr | 2018 | 2019 | 2020 | 2021 |
| EDGAR total | 264 | 267 | 276 | 280 |
| All sources (non-divergent $w$) | 171 | 184 | 177 | 188 |
| All sources (uncorrected $w$) | 153 | 175 | 152 | 133 |

The difference in change of emissions between GEOS-Chem simulation and TROPOMI is primarily due to the need to correct the final estimated emissions when using the original wind field. As the case (Fig. 2.) mentioned above, the final emission based on averaged daily divergence $(\overline{D_d^S})$ (Fig. 2d) apparently contains the residual of the divergence of background $(\overline{D_d^B})$ (Fig. 2c), which is highly correlated with the wind divergence $(\overline{D_d^W})$. However, this dependence is much smaller in the GEOS-Chem simulation and for the emissions derived from TROPOMI by using the non-divergent wind. Therefore, the correction applied to $\overline{D_d^S}$ to derive the final emission is also much

smaller. Considering the readability of the manuscript and the suggestion from the reviewer, we added in the revised manuscript from Line 232 to 249:

"Before we applied this change, we tested the non-divergent method in the GEOS-Chem simulation that was used in Liu et al., (2021). We found that this step slightly improved the capability of the method in resolving the spatial variability of sources (Figure S2), but underestimate the final emissions by about 15% in the GEOS-Chem simulation. In contrast, when deriving the emissions from TROPOMI, using a non-divergent wind field especially improves the robustness over coastal areas and typically increases emissions by 5-20% for most cases (Table S2 shows an example). The large changes occur in the western Turkmenistan and Iran & Iraq. Our previous method using the original wind data cannot distinguish sources from the high regional background, leading to the filtering of more potential sources and resulting in lower total emissions for an area. The difference in change of emissions between GEOS-Chem simulation and TROPOMI is primarily due to the correction of the final estimated emissions. As was mentioned in the manuscript, the final emission based on the divergence ($\overline{D_d^S}$). (Fig. 2d) apparently contains the residual of the divergence of background ($\overline{D_d^B}$) (Fig. 2c), which is highly correlated with wind divergence ($\overline{D_d^W}$). However, this dependence is much smaller for the GEOS-Chem simulation and for the emissions derived from TROPOMI by using non-divergent wind. The procedure and the evaluation of removing the wind divergence from the original wind field are explained in Part B in SI. Generally, using a non-divergent wind field can improve the capability of the method in resolving the sources, both in a model simulation and in TROPOMI observations."

[Figure]

**Figure S2.** The spatial distributions of (a) the average of a priori CH₄ emissions used in GEOS-Chem simulation, (b) corresponding estimated CH₄ emissions over June-August 2012 on a 0.625° lon. × 0.5° lat. grid in Liu et al., (2021). (c) The estimated CH₄ emissions by using the non-divergent wind. (d) The elevation map that is generated from GMTED2010 data set. (e) Scatter plots for emissions between a priori emissions estimated CH₄ emissions in (b) The black and green dots stand for the grid cells in the east [100°W-70°W, 25°N-48°N] and west [124°W-100°W, 25°N-48°N] of the domain.

**Specific comments**

(1) 170: The treatment of methane in EAC4 is not explained in Inness et al (2019), but from what I understand it uses a mass balance method to maintain the observed zonal mean background concentration. This means that there are "emissions" in the surface layer of the model to prevent the concentration from going down due to the atmospheric sink. Then what is mentioned here about only transport driven methane is incorrect. Even worse, the distribution of these emissions (or concentration corrections if you wish) does not resemble reality, which questions the realism of the simulated subcolumn above the PBL. But if the EAC4 column above PBL does not vary a lot on the spatial scales of interest this may not be much of an issue. The question then is if the method really benefits from turning total column into PBL columns. Wouldn't the method perform as well without subtracting EAC4 columns?

The EAC4 data is a daily reanalysis dataset designed for reactive trace gases such as NOₓ, SO₂, O₃ etc., with methane serving as a chemical background for reactions. Thus, it contains no a priori emissions of methane, and the observations of the global background sites are used to nudge simulation. The statement in the manuscript is indeed a bit misleading, so we changed the content at Line 193 to:

"This vertical column above the PBL, is based on the model results of EAC4 of CAMS at a relative high spatial resolution, 0.75° horizontally and 60 layers vertically (Inness et al., 2019), with methane serving as a background species for chemical reactions. This EAC4 model run contains no *a priori* $CH_4$ emissions. Thus, the spatial distribution of $CH_4$ is mainly driven by transport and orography, which will be subtracted from TROPOMI observations to estimate the PBL concentration of $CH_4$. It is important to note that the total dry air column from the EAC4 dataset is constrained by the TROPOMI retrieval for each pixel, which guarantees the mass conservation."

In Liu et al., (2021), we explained why converting the $XCH_4$ to $XCH_4$ in PBL is necessary. First, the enhancement due to transport in upper troposphere and stratosphere are irrelevant to surface emission but lead to a fake signal. Second, using horizontal winds at a certain height is not representative in calculating the divergence of $XCH_4$, which stands for the mean mixing ratio of the entire column, especially in areas with significant elevation changes. This can be illustrated by Figure S4 (referred to Figure A1 here) from SI of Liu et al., (2021). Fig. A1d and e are the divergence and corresponding estimated methane emission when using $XCH_4$ in troposphere. Large discrepancies occur over areas near the coast and at high latitudes. The uncertainty can be largely reduced by using the $XCH_4$ in PBL (Fig. A2b and c).

[Figure]

**Figure A1.** The spatial distributions of (a) the average of a priori $CH_4$ emissions used in GEOS-Chem simulation, (b) the divergence of CH4 sources in the PBL, and (c) corresponding estimated $CH_4$ emissions over June-August 2012 on a 0.625° lon. × 0.5° lat. grid. (d)-(e) are similar to (b)-(c) but for the results by using $XCH_4$ in the troposphere.

(2) l192: Besides divergence, orographic changes also influence XCH4 because they influence the weight of the stratospheric subcolumn - where methane mixing ratios are significantly lower. I do not see how this effect is accounted for in the method that it used. For a fixed PBL height above the surface, the EAC4 methane column should correlate with orography. I wonder if the variation in EAC4 is taken into account, and can actually be at the required spatial scale.

We answer this comment for two aspects. First, the reviewer asked if EAC4 takes the variation of the orography into consideration when simulating XCH4. Yes, although methane concentration is used as the chemical background, the simulation considers its dynamics. Figure A2a shows an example of 3-month average of XCH4 in 2020 from EAC4. We calculated the ratio of methane total column to total dry air density column at the similar overpassing time of TROPOMI in Fig. A2a. The spatial distribution of XCH4 reflects the orographic change (Fig. A2b) and latitude dependency. And it can be clearly seen that there is no a priori emission in EAC4 data when compared to TROPOMI observation (Fig. A2c).

Second aspect is about how we consider the orographic problem in practice. We should clarify here again that only the vertical profile of XCH4 from EAC4 is used in the divergence calculation of TROPOMI. The values of surface pressure, elevation, total dry air column, and total methane column of each pixel were taken from the WFMD product to ensure mass conservation. Here we use an example over Northern Iran (Figure A3), where the orography is quite complicated, to better explain how orographic effect is considered in our calculation. Apart from the enhancement due to the sources, the spatial distribution of TROPOMI observed XCH4 also show the feature of the orography (Fig. A3b and A3a). After we convert XCH4 to XCH4_PBL (Fig. A3c), the signal from the orography has been reduced. Although the remaining columns over some locations still contain the information of the elevation as the reviewer mentioned,

$\overline{D_d^B}$ can be used to diagnose the background residuals due to remained orographic effect

by equation (4): $E^{corr} = E' - (k \cdot \overline{D_d^B} + b)$. The biased high XCH4 value in PBL is

due to extremely low XCH4 used by EAC4 as a background. Actually, we had a detailed discussion about how we correct the residual of biased background due to orography in Liu et al., (2021) and in this paper from Line 183 to Line 187: "The advantages of including $X_d^B$ are (1) it can be used to diagnose the contribution of inhomogeneous background, especially over mountains and coastal regions, and (2) the system biases between CAMS and TROPOMI, which leads to biased $X_d^{PBL}$, is included in both and can be greatly reduced by subtracting $X_d^B$ from $X_d^{PBL}$."

[Figure]

**Figure A2.** (a) Elevation map on a grid of 0.75°, which is generated from the GMTED2010 data set at 30 arcsecs. Averaged XCH4 over January to March in 2020 based on (b) EAC4 dataset on a grid of 0.75° and (c) TROPOMI observation on a grid of 0.2°.

[Figure]

**Figure A3.** (a) Elevation map on a grid of 0.75°, which is generated from the GMTED2010 data set at 30 arcsecs. (b) TROPOMI averaged XCH4 and (b) XCH4 in 2019 on a grid of 0.2°.

(3) l198: The benefit of correcting w for divergence is not clear to me. The TROPOMI data have the imprint of the wind divergence, which the flux divergence method allows to elegantly account for. However, if you tweak the wind fields to remove divergence then I expect you end up projecting the TROPOMI observed impact of wind divergence on the surface fluxes. Or do you correct the TROPOMI data for this divergence component? If so, that should be explained better.

Please see the answer to the general comment why we correct for the non-divergent wind. Considering the readability of the manuscript and the suggestion from the reviewer, we mentioned in revised manuscript from Line 232 to 251:

"Before we applied this change, we tested the non-divergent method in the GEOS-Chem simulation that was used in Liu et al., (2021). We found that this step slightly improved the capability of the method in resolving the spatial variability of sources (Figure S2), but underestimate the final emissions by about 15% in the GEOS-Chem simulation. In contrast, when deriving the emissions from TROPOMI, using a non-divergent wind field especially improves the robustness over coastal areas and typically increases emissions by 5-20% for most cases (Table S2 shows an example). The large changes occur in the western Turkmenistan and Iran & Iraq. Our previous method using the original wind data cannot distinguish sources from the high regional background, leading to the filtering of more potential sources and resulting in lower total emissions

for an area. The difference in change of emissions between GEOS-Chem simulation and TROPOMI is primarily due to the correction of the final estimated emissions. As was mentioned in the manuscript, the final emission based on the divergence $(\overline{D_d^S})$. (Fig. 2d) apparently contains the residual of the divergence of background $(\overline{D_d^B})$ (Fig. 2c), which is highly correlated with wind divergence $(\overline{D_d^W})$. However, this dependence is much smaller for the GEOS-Chem simulation and for the emissions derived from TROPOMI by using non-divergent wind. The procedure and the evaluation of removing the wind divergence from the original wind field are explained in Part B in SI. Generally, using a non-divergent wind field can improve the capability of the method in resolving the sources, both in a model simulation and in TROPOMI observations."

(4) l285: I understand that by randomly selecting 80% of a time series a standard deviation can be computed, and that this standard deviation is larger when the time series is noisier. But what justifies using 80% to derive a presumably 1 sigma uncertainty range? Wouldn't it be better to take the standard deviation for individual days and divide by the square root of N or something like that? The errors in figure 5 look very optimistic to me, given the scatter plots in Liu et al, 2021 (for perfect winds and without measurement errors).

We apologize that the description about this part is quite confusing in the original manuscript, therefore we moved the detailed explanation from SI part B to the main text of revised manuscript from L332 to L365:

"The divergence method requires sufficient temporal records (typically more than 7 days with valid observation for a grid cell) to derive robust results. Thus, the divergence on a single day does not provide a realistic emission for that day, and the standard deviations for individual days does not reflect the uncertainty or variability of a source. In addition, this method is not suitable for sources with intermittent releases, such as sudden leaks in oil and gas production. $\overline{D_d^S}$ can be a quite large positive value for this kind of source. However, a small number of large releases in a time series may lead to removal of this source by the temporal filter (see the case of Fig. 6 in Sect. 4), which is built for automatically detecting retrieval artifacts over a large domain. In order to keep as many real sources as possible, we apply a Monte Carlo experiment to each possible source to estimate the uncertainty of the derived emissions and to evaluate the robustness/reliability of a source. The procedure is as follows:

(1) We randomly choose 80% of the sampling days from a time series in a year as a subset. We derive a new emission, $E_i$, and count the ratio, $R_i$, of the number of days that have larger normalized $D_d^S$ than normalized $D_d^B$.

(2) Repeat step (1) 30 times for a time series that has more than 20 sampling days while 10 times for the one that have fewer days to derive the set of emissions, $\{E_i\}$, and the set of ratios, $\{R_i\}$ for each possible source. $R_i$ is used as temporal filter in each sub-set.

(3) Take one-standard deviation of the set $\{E_i\}$ as an uncertainty of a source. If the median value ($R$) of $\{R_i\}$ is greater than 0.5, this source is regarded having high confidence, which means these emissions are constantly released and likely not caused by a retrieval artifact.

We also investigate the choice of the percentage of the time series and the number of the iterations. 80%-70% percent can be a reasonable range that ensure the representativeness as well as the randomness of sampling days. We have tested the number of iterations ranging from 10 to 50 times. The uncertainty map such as Fig. 5c typically become stable after 20 iterations, and 30 iterations can ensure the robustness as well as the efficiency of the calculation."

We found sources below the threshold of 3kg/km$^2$/h, also show large uncertainties (>20%). It implies they are sensitive to the regional background, and distinguishing them from the background is difficult.

(4) l318: But under low wind speed the XCH4 enhancement is much larger, and therefore easier detectable than at high wind speed. Then how do you relate a threshold XCH4 enhancement to a threshold emission enhancement?

Combined with the minimal enhancement that TROPOMI can resolve, as suggested by Hu et al. (2018) and Schneising et al., (2023), we estimate the detection threshold to be around 3kg/km$^2$/h according to the approach (section 5.1) in Jacob et al., (2022). This is, of course, a rough estimate since the threshold is quite variable with many factors such as the inversion method, temporal coverage, and spatial resolution. But as we mentioned above and in the original manuscript (L327-329): " However, we found that sources below the detection threshold show large uncertainties (>20%) in this study, which means the method is not robust to distinguish these small sources from the regional background".

Figure 5, caption: Averaged or total emissions?

Changed to "Averaged annual methane emissions derived from the divergence after the spatial correction". What we derived here contains no extra information of sectors, so it is total emission of all sectors.

Figure 5b: Is the arc above Riyadh real, or a remnant of the surface albedo related feature in figure 3?

They are not real sources but due to surface albedo in Figure 3. We change the title of Fig. 5b to "All possible sources above the threshold (3kg/km$^2$/h)."

Figure 7: The information about EMIT was in the caption of Figure 6 also. I advise to mention it once in the main text and remove it from the captions.

Changed. We added the introduction about EMIT instrument and the data in Sec. 2.2 "Methane bottom-up emission inventories and auxiliary emission datasets" at L152:

"To validate the sources not reported in bottom-up inventories, target-mode instruments with very high spatial resolution (pixels < 60m) (e.g., GHGSat, PRISMA, EMIT) are widely used to pinpoint individual sources and reveal their characteristics. NASA's Earth Surface Mineral Dust Source Investigation (EMIT) mission was launched in 2020 and methane plumes are recorded since 10[th] August 2022 (Source: https://earth.jpl.nasa.gov/emit/data/data-portal/Greenhouse-Gases/). It uses an advanced imaging spectrometer instrument that measures a spectrum for every point in the image. The high-confidence research grade methane plume complexes from point source emitters are released as they are identified (Brodrick et al., 2023)."

The relevant captions of Fig. 6 and 7 were removed.

l408: How do you know that the 5 selected regions are hardly influenced by retrieval issues?

It is indeed arbitrary to write in this way. We changed the content at L487 in our revised manuscript as follow:

"In Figure 8, we select five hotspot regions in the Middle East to further assess the annual regional emissions from 2019 to 2022. Before we calculate the emissions of each region, we checked spatial patterns of XCH4 and albedo from TROPOMI, as well as land features, to ensure no suspicious retrieval artifact is included as a source. The emissions are based on all possible sources and only confident sources are shown."

l422-423: Why not use the same filtering of 3km/km2/h for the inventory to quantify this difference?

There are two reasons that we keep all the sources in the bottom-up emission inventory. First, different bottom-up emission inventories can predict very different spatial distributions of methane sources, as an example we showed over Saudi Aribia, because the gridded data are typically converted from the country total emission by using some auxiliary data. Some small sources reported by one bottom-up emission inventory can be much lower than 3km/km$^2$/h but quite high in our estimated emission. A larger bias may occur if we exclude these small sources in different bottom-up inventories. Second, the threshold in this paper is decided by TROPOMI instrument and our algorithm, and

it does not represent the performance of the bottom-up emission inventory.

Figure 8: this figure would be easier to read if TROPOMI is one side of the bars and the inventories on the other side. Right row they are mixed. According to the color legend the pink bar is 'CEDS energy related CH4 emission' while according to the caption it is the total emission of sources > 3kg/km2/h.

Changed. We move TROPOMI derived emissions to one side. The caption is also corrected.

l428: more constant than what?

Change to "The preserved sources that pass the temporal filter are suggested to be more constant than other possible sources that did not pass the temporal filter."

l450: How do you know that if the emissions are constant the have a constant emission factor? I do not see how your method can separate between the emission factor and activity (which are multiplied in inventories to obtain the emission).

The constant source in our paper does not refer to one with a constant emission factor but indicates a source that continually releases methane for most days of a year. The temporal filter is built on the relatively magnitude of normalized $D_d^S$ and normalized $D_d^B$. For sources that are not affected by retrieval issues, the temporal filter acts more like a statistic, indicating how many days the source releases a distinct amount of methane that can be resolved by TROPOMI. Areas such as Tehran, Isfahan and the western coastal areas of Turkmenistan emit considerable amounts of methane on more than 50% of observation days, implying they constantly release the methane. We mentioned this in revised manuscript at L487:

"In Figure 8, we select five hotspot regions in the Middle East to further assess the annual regional emissions from 2019 to 2022. Before we calculate the emissions of each region, we checked spatial patterns of XCH4 and albedo from TROPOMI, as well as land features, to ensure no suspicious retrieval artifact is included as a source. The emissions are based on all possible sources and only confident sources are shown. The results of all possible sources (pink bars) may be more representative of the total emissions in these areas, and the emissions passing the temporal filters (blue bars) can be used to estimate the contribution of constant sources. Here we should clarify that the constant source in our paper does not refer to one with a constant emission factor but indicates a source that continually releases methane for most days of a year."

TECHNICAL CORRECTIONS

l145: asses io access

Changed.

Changed.

Changed to "or is caused".

Changed to "we found that sources".

Changed to "is aim at identifying".

Changed to "metal".

Changed.

l317: 'explanation' io 'explanations'

Changed.

Changed.

Figure 6, caption: 'observations in 2019', 'the EMIT instrument', 'the VIRRS instrument'

Changed.

Figure 7, caption: see my comments about figure 6

Changed.

Changed.